# Cord Blood Advanced Lipoprotein Testing Reveals an Interaction between Gestational Diabetes and Birth-Weight and Suggests a New Early Biomarker of Infant Obesity

**DOI:** 10.3390/biomedicines10051033

**Published:** 2022-04-29

**Authors:** Francisco Algaba-Chueca, Elsa Maymó-Masip, Mónica Ballesteros, Albert Guarque, Alejandro Majali-Martínez, Olga Freixes, Núria Amigó, Sonia Fernández-Veledo, Joan Vendrell, Ana Megía

**Affiliations:** 1Department of Endocrinology and Nutrition and Research Unit, Hospital Universitari de Tarragona Joan XXIII, Institut d’Investigació Sanitària Pere Virgili (IISPV), Dr. Mallafre Guasch, 4, 43005 Tarragona, Spain; falgabachueca@gmail.com (F.A.-C.); elsamaymomasip@gmail.com (E.M.-M.); ofreixess.hj23.ics@gencat.cat (O.F.); sonia.fernandezveledo@gmail.com (S.F.-V.); 2CIBER de Diabetes y Enfermedades Metabólicas Asociadas (CIBERDEM)—Instituto de Salud Carlos III, 28029 Madrid, Spain; nuriaamigo@gmail.com; 3Departament of Basic Medical Sciences and Department of Medicine and Surgery, Rovira i Virgili University, 43005 Tarragona, Spain; ballesterosperez.monica@gmail.com (M.B.); albertguarque@gmail.com (A.G.); 4Department of Obstetrics and Gynecology, Hospital Universitari de Tarragona Joan XXIII, Institut d’Investigació Sanitària Pere Virgili (IISPV), Dr. Mallafre Guasch, 4, 43005 Tarragona, Spain; 5Department of Obstetrics and Gynecology, Medical University of Graz, 8036 Graz, Austria; a.majali-martinez@medunigraz.at; 6Biosfer Teslab SL Plaça del Prim, 10 2on 5a, 43201 Reus, Spain

**Keywords:** birth-weight, gestational diabetes, lipoprotein profile, obesity, fetal blood

## Abstract

Abnormal lipid metabolism is associated with gestational diabetes mellitus (GDM) and is observed in neonates with abnormal fetal growth. However, the underlying specific changes in the lipoprotein profile remain poorly understood. Thus, in the present study we used a novel nuclear magnetic resonance (NMR)-based approach to profile the umbilical cord serum lipoproteins. Two-dimensional diffusion-ordered 1H-NMR spectroscopy showed that size, lipid content, number and concentration of particles within their subclasses were similar between offspring born to control (*n* = 74) and GDM (*n* = 62) mothers. Subsequent data stratification according to newborn birth-weight categories, i.e., small (*n* = 39), appropriate (*n* = 50) or large (*n* = 49) for gestational age (SGA, AGA and LGA, respectively), showed an interaction between GDM and birth-weight categories for intermediate-density lipoproteins (IDL)-cholesterol content and IDL- and low-density lipoproteins (LDL)-triglyceride content, and the number of medium very low-density lipoproteins (VLDL) and LDL particles specifically in AGA neonates. Moreover, in a 2-year follow-up study, we observed that small LDL particles were independently associated with offspring obesity at 2 years (*n* = 103). Collectively, our data demonstrate that GDM disturbs triglyceride and cholesterol lipoprotein content across birth-weight categories, with AGA neonates born to GDM mothers displaying a profile more similar to that of adults with dyslipidemia. Furthermore, an altered fetal lipoprotein pattern was associated with the development of obesity at 2 years.

## 1. Introduction

Fetal growth and development constitute a particularly vulnerable period in life that is greatly affected by the maternal environment. Prenatal exposure to nutritional stressors has been associated with fetal programming, which can impact both metabolism and physiology and, consequently, predispose to later development of cardiovascular and metabolic diseases, including obesity [1]. In this context, it has been proposed that cardiovascular disease can begin early in life [2] and that atherosclerosis may originate during the fetal period [3].

Birth-weight is strongly determined by neonatal fat mass and gestational age, and fetal growth disorders can result from impaired maternal and fetal lipid metabolism. In fact, the levels and composition of cord blood lipids, apolipoproteins and lipoproteins are affected by both maternal and fetal factors [4,5,6]. Disturbed lipid profiles at birth have been described in small and large for gestational age (SGA and LGA, respectively) neonates [6,7,8,9]. When compared with appropriate for gestational age (AGA) peers, SGA neonates show higher levels of triglycerides, triglyceride-enriched very low-density lipoproteins (VLDL), low-density lipoproteins (LDL) and high-density lipoproteins (HDL), and lower levels of total cholesterol [6,9,10]. By contrast, LGA neonates display higher LDL, HDL and total cholesterol levels than AGA neonates [8].

Diabetic pregnancies are associated with a higher incidence of fetal growth disorders, and there is evidence that disturbances in maternal metabolism strongly contribute to these situations [11]. Changes in cord blood lipoprotein concentrations have been reported in mothers with type 1 diabetes mellitus, including an increased cholesterol content of LDL and a decrease in HDL [12,13]. The situation appears more complex in gestational diabetes mellitus (GDM), with some studies showing no differences with normal glucose-tolerant mothers and others showing lower HDL and higher VLDL and LDL cholesterol concentrations [14]. Additionally, qualitative changes in HDL remodeling resulting in an altered functionality have been reported in GDM neonates [15], but that does not seem to affect newborn cholesterol metabolism in both obese and well-controlled GDM mothers [16]. Thus, new strategies for an in-depth analysis of the lipoprotein profile in GDM are still required.

Interestingly, 1H-nuclear magnetic resonance (1H-NMR)-based analysis of lipoproteins has established that the number of LDL and HDL particles is a more powerful index of cardiovascular risk than classical cholesterol determinations, given the large variability in the amount of cholesterol per particle and in particle size [17]. 1H-NMR-based tests have also demonstrated the incomplete conversion of VLDL into LDL in diabetes, which results in a higher prevalence of VLDL and small and dense LDL particles [17].

Given the heterogeneity of growth patterns and the inconclusive findings in the cord blood lipoprotein profile of infants from GDM mothers, a comprehensive characterization of the main lipoproteins, including the assessment of the size and number of particles, is necessary to identify possible alterations in fetal lipoprotein metabolism and their potential consequences for fetal health. Thus, in the present study, we used the Liposcale test, a novel advanced lipoprotein assessment method based on 2D diffusion-ordered 1H-NMR [18], to examine possible correlations between fetal growth disorders and differences in umbilical cord blood lipoprotein profile. Additionally, we explored the potential association with offspring outcomes, including obesity, at 2 years of age.

## 2. Materials and Methods

### 2.1. Study Subjects

All the mother–offspring pairs included in this study belonged to a carefully selected pre-birth cohort, and the mothers gave birth at the Department of Obstetrics of the Hospital Universitari de Tarragona Joan XXIII between June 2010 and May 2017. GDM and control women were recruited at their first prenatal visit to the Obstetrics Department and were followed until delivery. In this study, the participants were stratified according to the birth-weight category, and we included a similar proportion of infants born to mothers with GDM and control women who were AGA, SGA or LGA. The inclusion criteria at the end of pregnancy were: (1) singleton pregnancy, (2) accurate gestational age confirmed by an ultrasound examination before 20 weeks of gestation, (3) absence of fetal anomalies, (4) cord blood serum availability and (5) maternal lipid information from the third trimester exam. Neonates with major congenital anomalies, intrauterine infections or born from women with chronic and inflammatory diseases were excluded. One hundred and thirty-six neonate-and-mother pairs fulfilled these criteria (74 control pairs and 62 pairs with GDM) and were included in this study, which was performed in accordance with the tenets of the Declaration of Helsinki and whose protocol was reviewed and approved by the Hospital Universitari de Tarragona Research Ethics Board (ref: 243/2016). All participants provided informed consent before inclusion.

All mothers were screened for GDM between 24–28 weeks of pregnancy following the Spanish Diabetes and Pregnancy Group recommendations [19]. Subjects with a 1 h 50 g glucose challenge test ≥140 mg/dL underwent a 3 h 100 g oral glucose tolerance test. Subjects with two or more values above the threshold proposed by the National Diabetes Data Group [20] were considered to have GDM, whereas those with all values below the threshold were classified as controls.

Care for GDM was managed according to the Spanish guidelines for diagnosis and therapy of GDM [19]. Women with GDM were given an individualized diet with at least 40% carbohydrates, and they were instructed to self-monitor blood glucose 6 times a day (fasting and 1 h postprandial). Insulin therapy was recommended when fasting glucose and/or 1 h post-prandial values were repeatedly over 95 mg/dL (5.3 mmol/L) or over 140 mg/dL (7.8 mmol/L), respectively. According to these criteria, 31 women were treated only with diet, and 31 women also required insulin.

Maternal and umbilical cord blood samples were stored in a biobank collection along with the associated clinical data.

### 2.2. Clinical and Demographic Data

Demographic and obstetric information on participants was collected via an interviewer-administered questionnaire, which paid particular attention to GDM risk factors. Maternal anthropometry included height, pre-pregnancy weight and weight at the end of pregnancy. Pre-pregnancy weight was self-reported and compared with the weight recorded in the first prenatal visit (before the 10th week of pregnancy) to ensure concordance. Pre-pregnancy body mass index (BMI) and final BMI were calculated as pre-pregnancy weight (kg)/height (m)^2^ and final weight/height (m)^2^, respectively. Similarly, gestational weight gain (GWG) was calculated as final weight—pre-pregnancy weight.

Infant data included sex, gestational age, method of delivery and anthropometry. Neonatal length and weight were measured after delivery using a measuring board to the nearest 0.1 cm and a calibrated scale to the nearest 10 g. Ponderal index (PI) was calculated as birth weight (g)/length (cm)^3^. Suprailiac skinfold thickness was measured within the first 48 h of life and was used to calculate the fat mass percentage. [21] Neonates were classified according to gestational age- and sex-specific growth charts of the World Health Organization (WHO) [22]. Infants with birth weights adjusted for sex and gestational age below the 10th percentile were considered SGA, while those with birth weights above the 90th percentile were included in the LGA group. Infants with birth weights between the 10th and 90th percentile adjusted for gestational age were included in the AGA group. The distribution of neonates according to birth-weight category was: 25 AGA, 25 SGA and 24 LGA in the control group and 25 AGA, 14 SGA and 23 LGA in the GDM group.

### 2.3. Infant Growth and Child BMI

Height and weight information from birth up to 2 years of age was collected for 103 children. We defined obesity as a BMI ≥ 85th percentile according to age- and sex-specific BMI tables of the WHO growth standards [22].

### 2.4. Umbilical Cord Blood Collection

Umbilical cord blood was obtained immediately after delivery. Serum was immediately separated by centrifugation, divided into aliquots and stored at −80 °C until further analysis.

### 2.5. Laboratory Analysis

Maternal fasting serum samples were obtained between gestational week 33 and 36 to determine glucose, triglycerides, total and HDL-cholesterol in an ADVIA 2400 (Siemens AG, Munich, Germany) autoanalyzer by standard enzymatic methods [23]. LDL cholesterol was calculated using the Friedewald formula. Plasma insulin was determined by immunoassay in an ADVIA Centaur System (Siemens AG, Munich, Germany). This assay shows a cross-reactivity of 0.1% to intact human proinsulin and the primary circulating split form des-31,32-proinsulin. Insulin resistance was estimated using homeostatic model assessment of insulin resistance (HOMA)-IR, as previously described [24].

### 2.6. 1H-NMR Spectroscopy-Based Cord Blood Lipoprotein Profiling

Cord blood serum samples were analyzed using the 2D diffusion-ordered 1H-NMR-based Liposcale test (Biosfer Teslab, Reus, Spain) [18]. This technique has shown to be reliable with samples stored at −80 °C for more than a decade [25]. The test provides information about size, lipid concentration (cholesterol and triglycerides), number of particles and concentration of particles within their subclasses (large, medium and small) for the main classes of lipoproteins, i.e., VLDL, LDL, intermediate-density lipoprotein (IDL) and HDL. 1H-NMR spectra were recorded on a Bruker Avance III 600 spectrometer (Bruker BioSpin, Rheinstetten, Germany).

### 2.7. Statistical Analysis

SPSS software v20.0 (IBM, Armonk, NY, USA) was used for statistical analysis. Data were presented as percentages for categorical variables, mean (±SD) for normally distributed continuous variables, and median (interquartile range) for non-normally distributed variables. Normal distribution of the data was tested with the Kolmogorov–Smirnov test. Non-normally distributed quantitative variables were used after log10 transformation, when required. For comparisons of proportions, differences between groups were analyzed using the chi-square test, while comparisons between normally and non-normally distributed quantitative variables were performed using unpaired *t*-test or Mann–Whitney U test. One-way analysis of variance (ANOVA) was used to test differences among three or more groups. Potential interactions between GDM and birth-weight categories were assessed by two-way ANOVA followed by Bonferroni post hoc test to adjust for multiple comparisons. Spearman’s rank correlation coefficients were used for the analysis of the relationships between 1H-NMR-assessed lipoprotein profile and maternal and offspring metabolic and clinical variables. To control the false discovery rate (FDR), the Benjamini–Hochberg (B–H) procedure was used, and only those values significant with the B–H correction were considered [26]. Logistic regression was used to investigate the independence of the association between 1H-NMR-assessed large LDL and small LDL particles, offspring obesity (percentile ≥ 85th = 1), and the normal weight (percentile < 85th = 0), after adjustment for potential confounders (GDM, gestational age at delivery, birth weight, sex, pre-gestational BMI and gestational weight gain). *p* < 0.05 was considered statistically significant.

## 3. Results

### 3.1. Clinical Characteristics and Cord Blood 1H-NMR-Based Lipoprotein Profile of the Studied Population

Clinical and metabolic characteristics of the two groups, i.e., control and GDM, are shown in Table 1. Clinical and laboratory parameters and the 1H-NMR lipoprotein profile were similar between GDM and control groups with the exception of GWG, which was significantly lower in the GDM group (Table 1). In the control group, 42 pregnant women were normal weight, 18 overweight and 14 obese. In the GDM group, 27 pregnant women were normal weight, 20 overweight and 15 obese. Likewise, GWG (7.2 ± 4.4 vs. 10.1 ± 5.1 kg; *p* = 0.023) and fasting glucose (78 ± 8 vs. 89 ± 12 mg/dL; *p* < 0.001) were lower in the GDM group of women treated only with diet compared to those who needed insulin therapy (data not shown). In the 1HNMR-assessed lipoprotein profile, no difference was observed between the two groups, except for a lower concentration of medium HDL-P in GDM treated with insulin compared with those treated with diet (10.3 ± 1.2 vs. 9.6 ± 1.4; *p* = 0.026) (data not shown).

One hundred and thirty-six neonates born to GDM (*n* = 62) and control (*n* = 74) mothers were categorized into three groups based on birth-weight categories according to age- and sex-weight specific charts (Table 2). As expected, there were significant differences in birth weight, percentage of fat mass and PI across groups, increasing from the SGA to the LGA group. There were also differences between birth-weight groups for GWG, final BMI and cord blood insulin. To note, in the GDM group the type of intervention (only diet or diet plus insulin) was distributed similarly in the three birth-weight groups (*p* = 0.321) (data not shown).

In the 1HNMR-assessed cord blood lipoprotein profile, the cholesterol content in VLDL, LDL and HDL lipoproteins was different between the three birth-weight groups, with the LGA and AGA groups showing the highest cholesterol content in VLDL and HDL lipoproteins, respectively, and the SGA group showing the lowest cholesterol content in LDL and HDL lipoproteins. Regarding triglyceride content, SGA showed a higher VLDL triglyceride content than AGA and LGA, whereas AGA showed higher HDL triglyceride content than SGA and LGA. No differences were observed in LDL lipoproteins (Table 2).

Lipoprotein particle size and number were also different across the three groups. The number of VLDL-particles (VLDL-P) was highest in the SGA group and lowest in the LGA group, which was consistent with the differences observed among particle sizes (large, medium and small VLDL-P). The number of LDL-P was lower in the SGA group than in the AGA and LGA groups, and the same distribution was observed for large and small LDL-P. The number of HDL-P was highest in the AGA group and showed an inverse U distribution when compared with the LGA and SGA groups. This phenomenon was observed specifically for small particles (Table 2).

### 3.2. GDM Alters the Cord Blood Lipoprotein Profile across Birth-Weight Categories

While the cord blood lipoprotein profile was similar in offspring born to GDM and control mothers, we found some interactions when we assessed the effect of both birth-weight categories and GDM. AGA neonates born to GDM mothers had higher IDL-cholesterol and -triglyceride content, and higher LDL-triglyceride content than the SGA and LGA groups, in contrast to those born to control mothers who had lower concentrations. The same pattern was also observed with medium VLDL-P and LDL-P, which followed an inversed U distribution (Figure 1 and Table 3).

### 3.3. Relationship of 1H-NMR-Assessed Lipoprotein Profile with Clinical and Laboratory Parameters

Subsequently, we examined the relationship between the 1H-NMR-based lipoprotein profile and the maternal clinical and laboratory parameters as well as the neonatal outcomes. This analysis was conducted separately in both GDM and control groups, and results are shown in Figure 2. In the GDM group, maternal LDL-cholesterol determined by standard methods was negatively associated with cord blood VLDL-P (total number, large and small particles), cholesterol and triglyceride content in VLDL, and IDL- and HDL-triglyceride content.

LDL-cholesterol concentrations were negatively associated with birth weight in the GDM group, while no association was observed in the control group. Cord blood insulin was strongly and positively associated with small and large LDL-P and LDL-cholesterol content in the control group, while a negative relationship was observed with VLDL-P and VLDL-triglyceride content in the GDM offspring.

In both groups, neonatal adiposity was negatively correlated with cord blood VLDL-P, VLDL- and IDL-triglyceride content, and positively associated with cord blood LDL-cholesterol content and LDL-P, and these associations were stronger in the control group.

### 3.4. Cord Blood 1H-NMR Lipoprotein Profile Is Associated with Obesity at Two Years

To assess whether the 1H-NMR-determined cord blood lipoprotein profile could be used as a biomarker for offspring outcomes, we explored its potential association with obesity at 2 years of life in a subset of participants. From the 103 children available for the follow-up study, 78 had a normal weight and 25 were obese. Obese children were born to women with higher pregestational BMI (25.6 ± 5.0 vs. 28.2 ± 6.7 kg/m^2^; *p* = 0.036), had higher birth weight (3281 ± 609 vs. 3632 ± 480 g; *p* = 0.010) and were more exposed to GDM during the intrauterine life compared with normal-weight children (17 of the 25 obese children and 34 of the 78 normal-weight children were born to a mother with GDM; *p* = 0.034). Additionally, they showed a higher number of cord blood small (352 ± 29 vs. 326 ± 51 nmol/L; *p* = 0.019) and large LDL-P (98 ± 11 vs. 91 ± 13 nmol/L; *p* = 0.022). No other differences were observed between the two groups. To further assess the independence of these associations, we performed logistic regression analysis. We found that small LDL-P were associated with infant obesity at 2 years after adjusting for potential confounders (*p* = 0.023), whereas large LDL-P showed a trend (*p* = 0.058) (Table 4). No differences in sex, GDM or birth-weight category distribution, birth weight, maternal pre-pregnancy BMI, GWG or cord blood lipid profile were observed between the children that could not be followed up and those that remained in the study.

## 4. Discussion

Both GDM and abnormal growth patterns have been associated with long-term adverse outcomes in offspring. Similarly, changes in the lipoprotein composition have been proposed as potential markers of cardiovascular diseases later in life. Taking advantage of a thorough lipoprotein profiling based on 1H-NMR, we showed for the first time that GDM modifies the umbilical cord blood lipoprotein profile in AGA neonates. In particular, GDM alters IDL lipoproteins, triglyceride content in LDL, and medium-size VLDL-P and LDL-P in those children. By contrast, GDM offspring belonging to the LGA and SGA groups have lipoprotein profiles more similar to those of controls. Additionally, we found that cord blood small LDL-P, known to be associated with atherosclerosis development, had predictive value for later obesity in the offspring.

Both under and overnutrition in utero affects the lipoprotein profile of neonates [4,5,6,7]. SGA neonates are reported to exhibit higher cord blood triglyceride concentrations [6,9,27], higher VLDL and IDL concentrations, and lower HDL concentrations when compared with equivalent AGA neonates [10]. Some of these findings have also been reported in fetal macrosomia [6,8] and GDM pregnancies [28]. However, standard lipid profiling failed to identify differences between offspring born to healthy pregnant and GDM mothers [29]. Nonetheless, as shown in other metabolic disorders such as diabetic dyslipidemia [17], a more in-depth characterization of the lipoprotein profile could provide more accurate data on the regulation of lipoprotein metabolism in fetal life and its potential implications for metabolism later in life. Thus, using 1H-NMR-based cord blood lipoprotein profiling, we detected differences according to fetal growth categories in GDM women, revealing a disturbed cholesterol and triglyceride metabolism predominantly in AGA neonates. This pattern may denote an excessive transfer of triglycerides to LDL, and the further increased cholesterol-poor LDL particles in the liver [30]. Furthermore, nutritional factors and dysfunctional HDL lipoproteins [31], as described in cord blood of infants of GDM mothers [15], may induce an abnormal hepatic lipase activation, also increasing IDL half-life. This scenario is similar to the dyslipidemia associated with diabetes and insulin-resistant states, where an increased generation of IDL, small and dense LDL particles, and triglyceride-enriched HDL particles is observed [30], and which has been related to an increased atherogenic risk. These findings appear to suggest that postnatal insulin resistance, which has been described in offspring of GDM women, may be programmed in utero and would be present even in AGA neonates, further suggesting that good glycemic control during pregnancy is not enough to prevent long-term complications, as has been previously reported [32,33].

Since treatment with insulin at the end of pregnancy may activate placental nutrient transport to the fetus and promote placental fatty acid transfer [34], we assessed potential differences in the advanced lipoprotein profile and the distribution of infants between the two groups in case it could be a confounding factor. No differences in the 1HNMR-assessed lipoprotein profiles between GDM women treated with diet or insulin were noticed, except for a lower concentration of medium HDL-P in the insulin-treated group, as we have stated in the results section. Additionally, a similar distribution according to birth-weight categories was found between the diet- and insulin-treated groups. For this reason, they were analyzed together.

Despite the differences observed in some lipoproteins and lipid particles in AGA infants of GDM women and controls, these differences were not seen in the rest of the lipoprotein particles or in the groups characterized by more severe alterations in growth, SGA and LGA, suggesting that in these cases, the pattern of growth and fat accumulation could dilute the effect produced by GDM. However, this has to be interpreted cautiously since evidence of cord blood lipoproteins as biomarkers for cardiovascular disease later in life is still scarce. In this regard, results in adult life are often extrapolated to fetal life, despite the fact that they might have a different interpretation. In fact, differences in lipoprotein composition between adults and fetuses have been described, including excess apoE on fetal HDL particles, which are large in size, lack paraoxonase I, and might have lower anti-oxidant capacity [35,36], as well as small LDL poorer in lipid content [37]. These findings highlight the need for a better understanding of how lipid metabolism in utero relates to lipid metabolism in adults and, in turn, how these metabolic changes in the fetus impact adult cardiovascular health.

Previous studies exploring the potential relationship between prenatal lipid metabolism and adverse metabolic outcomes in offspring have generated inconsistent results [38,39,40,41,42,43]. Following other reports [44,45], we confirmed that GDM, pre-pregnancy BMI, and GWG during pregnancy are all associated with offspring obesity in early life. Furthermore, we found that small LDL-P in cord blood were associated with early obesity, even after controlling for confounding factors. These findings support the notion that disturbances in lipoprotein metabolism at birth may have lasting effects independently of birth weight or maternal metabolic status.

There is evidence that an altered fetal lipoprotein profile is associated with aorta intima thickness in SGA and LGA neonates [8,27], indicating a potentially increased atherosclerotic risk already at birth. We are aware that our results cannot establish a direct link between the 1H-NMR-assessed lipoprotein profile, observed in GDM-AGA newborns, and a potentially increased atherogenic risk. Nevertheless, the present study offers new clues to understand the high metabolic and cardiovascular risk in the offspring of pregnant GDM women [46]. Long-term studies are guaranteed to confirm whether cord blood 1H-NMR-based lipoprotein profiling can be implemented as a useful biomarker of later metabolic diseases beyond 2 years of age.

One of the main limitations in observational studies is the inability to attribute causation between the observed associations. However, we considered several critical confounding variables to mitigate bias in the analysis. Thus, the main prenatal factors were addressed, and the groups were comparable for maternal BMI and birth-weight categories. Of note, to reach a sufficient sample size in the three birth-weight categories, the SGA and LGA groups were overrepresented, and further population-based studies are needed to determine the role of lipoprotein composition and subfractions in the pathogenesis of metabolic diseases in offspring. Furthermore, given the relevance of the placenta in the passage of nutrients and in the regulation of fetal metabolism and growth, it would be important to establish whether there is a parallelism between the lipoprotein pattern and placental metabolic pathways.

The strengths of this study include a longitudinal birth cohort with almost complete maternal data that establish a temporal relationship between the outcome and the exposure to GDM. The novelty of the lipoprotein assessment, which allowed us to identify different fetal metabolic behaviors, is also a big asset in the experimental methods.

## 5. Conclusions

GDM disturbs triglyceride and cholesterol lipoprotein concentrations across birth categories, with GDM-AGA neonates showing a profile more similar to that of adults with dyslipidemia and atherosclerosis than those born to normal glucose-tolerant mothers. Moreover, an altered fetal lipoprotein pattern is associated with obesity development at 2 years. Overall, these findings suggest that the fetal lipoprotein profile might be an early biomarker for the development of later diseases.

## Figures and Tables

**Figure 1 biomedicines-10-01033-f001:**
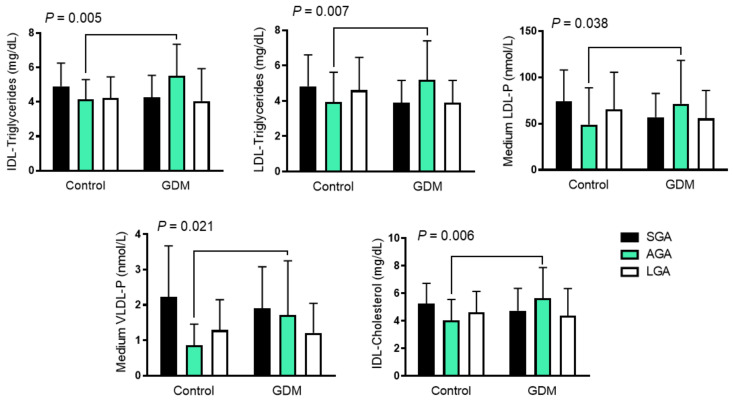
Differences in cord blood 1H-NMR-assessed lipoprotein pattern among growth groups in GDM and control mothers. Data are shown as mean ± SD and were analyzed using two-way ANOVA. GDM: gestational diabetes mellitus; LDL low-density lipoprotein; VLDL: very-low-density lipoprotein; IDL: intermediate-density lipoproteins; IDL-C: cholesterol content in IDL; IDL-TG: Triglyceride content in IDL; LDL-TG: triglyceride content in LDL; LDL-P: LDL number of particles; VLDL-P: VLDL number of particles; AGA: appropriate for gestational age; LGA: large for gestational age, SGA; small for gestational age.

**Figure 2 biomedicines-10-01033-f002:**
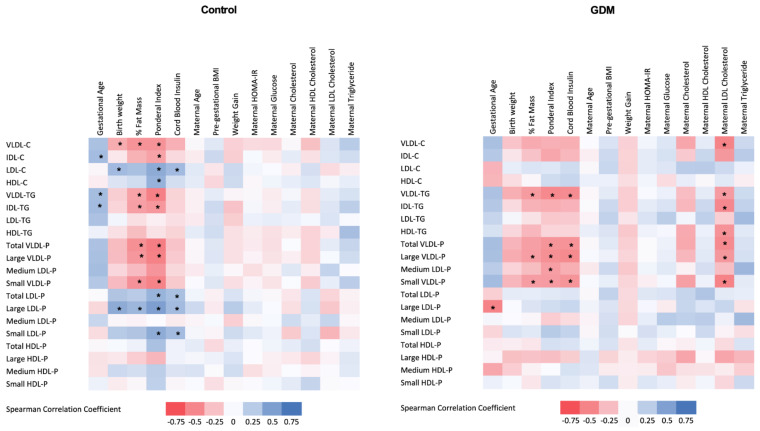
Heat map of the associations between cord blood 1H-NMR-assessed lipoprotein profile and maternal and neonatal clinical variables in control (left panel) and GDM (right panel) mothers. GDM: gestational diabetes mellitus; % Fat Mass: percentage of fat mass; Cb: cord blood; HDL: high-density lipoprotein; LDL low-density lipoprotein; VLDL: very-low-density lipoprotein; IDL: intermediate-density lipoprotein; VLDL-C: cholesterol content in VLDL; VLDL-TG: triglyceride content in VLDL; VLDL-P: VLDL number of particles; IDL-C: cholesterol content in IDL; IDL-TG: triglyceride content in IDL; LDL-C: cholesterol content in LDL; LDL-TG: triglyceride content in LDL; LDL-P: LDL number of particles; HDL-C: cholesterol content in HDL; HDL-TG: triglyceride content in HDL; HDL-P: HDL number of particles; Pre-pregnancy BMI: pre-gestational body mass index; HOMA-IR: homeostatic model assessment for insulin resistance. Spearman correlation coefficients. * Indicates significant associations after applying B–H procedure for FDR correction.

**Table 1 biomedicines-10-01033-t001:** Maternal and neonatal characteristics and cord blood 1H-NMR-assessed lipoprotein profile according to gestational diabetes.

**Maternal and Neonatal Clinical Characteristics**			
	**Control (*n* = 74)**	**GDM (*n* = 62)**	***p*-Value**
Maternal age (years)	32.5 ± 5.4	33.5 ± 4.3	0.257
Pre-gestational BMI (kg/m^2^)	25.5 ± 5.2	26.6 ± 5.1	0.207
Gestational weight gain (kg)	12.3 ± 6.2	8.6 ± 4.9	<0.001
Final BMI (kg/m^2^)	30.1 ± 4.8	30.0 ± 4.7	0.872
Smoking, *n* (%)	14 (18.9)	8 (12.9)	0.363
M cholesterol (mg/dL) *	246 ± 40	236 ± 44	0.165
M HDL cholesterol (mg/dL) *	73 ± 13	72 ± 15	0.530
M LDL cholesterol (mg/dL) *	121 ± 54	124 ± 36	0.775
M triglycerides (mg/dL) *	205 ± 79	203 ± 81	0.898
HOMA-IR *	2.0 (1.2–3.3)	2.9 (1.66–4.22)	0.052
Gestational age (weeks)	39 (38–40)	39 (38–40)	0.749
Vaginal delivery *n* (%)	51 (68.9)	50 (80.6)	0.119
Birth weight (g)	3259 ± 603	3310 ± 697	0.645
Male sex *n* (%)	38 (51.4)	32 (51.6)	0.976
SGA/AGA/LGA (*n*)	25/25/24	14/25/23	0.353
Fat mass (%)	11.7 ± 4.3	11.9 ± 3.8	0.789
Ponderal Index (g/cm^3^)	2.7 ± 0.3	2.7 ± 0.3	0.633
Cord blood insulin (mcUI/mL)	4.5 (2.1–8.0)	6.3 (2.9–12.1)	0.058
**Cord blood 1H-NMR-assessed lipoprotein profile**			
	**Control (*n* = 74)**	**GDM (*n* = 62)**	***p*-value**
VLDL-cholesterol (mg/dL)	7.2 ± 3.3	7.6 ± 4.0	0.495
IDL-cholesterol (mg/dL)	4.6 ± 1.6	5.0 ± 2.0	0.291
LDL-cholesterol (mg/dL)	70.1 ± 10.1	70.6 ± 8.7	0.767
HDL-cholesterol (mg/dL)	40.9 ± 8.8	40.6 ± 8.1	0.827
VLDL-triglycerides (mg/dL)	29.4 ± 8.6	30.7 ± 9.9	0.415
IDL-triglycerides (mg/dL)	4.4 ± 1.3	4.7 ± 1.8	0.335
LDL-triglycerides (mg/dL)	4.5 ± 1.8	4.4 ± 1.9	0.925
HDL-triglycerides (mg/dL)	7.9 ± 3.7	8.6 ± 3.9	0.251
VLDL-P (nmol/L)	23.1 ± 6.3	24.0 ± 7.5	0.416
Large VLDL-P (nmol/L)	0.6 (0.5–0.8)	0.6 (0.5–0.8)	0.848
Medium VLDL-P (nmol/L)	1.1 (0.6–2.0)	1.1 (0.7–2.4)	0.471
Small VLDL-P (nmol/L)	21.0 ± 5.1	21.8 ± 6.3	0.408
LDL-P (nmol/L)	484.9 ± 71.7	488.0 ± 62.0	0.79
Large LDL-P (nmol/L)	92.5 ± 13.0	93.1 ± 11.0	0.782
Medium LDL-P (nmol/L)	61.9 ± 39.3	62.1 ± 36.9	0.973
Small LDL-P (nmol/L)	330.4 ± 50.1	332.7 ± 42.1	0.775
HDL-P (nmol/L)	17.8 ± 4.2	17.4 ± 4.1	0.591
Large HDL-P (nmol/L)	0.3 ± 0.1	0.3 ± 0.1	0.107
Medium HDL-P (nmol/L)	9.7 ± 1.5	9.9 ± 1.4	0.257
Small HDL-P (nmol/L)	7.8 ± 3.8	7.1 ± 3.9	0.306

Data are presented as mean ± SD and median (IQR, 25–75) for parametric and non-parametric variables, respectively. Statistical analysis included *t*-test and the Mann–Whitney U test. GDM: gestational diabetes mellitus; M: maternal blood; BMI: body mass index; HOMA-IR: Homeostatic Model Assessment for Insulin Resistance; LDL: low-density lipoproteins; HDL: high-density lipoproteins; VLDL: very low-density lipoproteins; IDL: intermediate-density lipoproteins; VLDL-P: VLDL number of particles; IDL-P: IDL number of particles; LDL-P: LDL number of particles; HDL-P: HDL number of particles; SGA: small for gestational age, AGA: appropriate for gestational age; LGA: large for gestational age. * Parameters measured in maternal blood obtained in gestational weeks 33–36. Only 70 control and 51 GDM women were available for the analysis.

**Table 2 biomedicines-10-01033-t002:** Maternal and neonatal characteristics and cord blood 1H-NMR-assessed lipoprotein profile according to birth-weight groups.

**Maternal and Neonatal Clinical Characteristics**				
	**SGA (*n* = 39)**	**AGA (*n* = 50)**	**LGA (*n* = 49)**	***p*-Value**
Maternal age (years)	31.6 ± 4.6	33.91 ± 5.0	32.8 ± 5.0	0.102
Pre-gestational BMI (kg/m^2^)	25.3 ± 4.2	25.9 ± 5.3	26.7 ± 5.7	0.445
Gestational weight gain (kg)	8.6 ± 5.0	10.0 ± 5.3	12.9 ± 6.3 ^b^	0.002
Final BMI (kg/m^2^)	28.6 ± 4.0 ^b^	29.7 ± 4.6	30.0 ± 4.8	0.014
Smoking, *n* (%)	11 (28.2)	5 (10.6)	6 (12)	0.053
M cholesterol (mg/dL) *	247 ± 36	235 ± 46	244 ± 42	0.395
M HDL cholesterol (mg/dL) *	73 ± 10	76 ± 17	69 ± 12	0.053
M LDL cholesterol (mg/dL) *	130 ± 47	116 ± 35	124 ± 57	0.439
M triglycerides (mg/dL) *	191 ± 70	204 ± 79	209 ± 74	0.552
HOMA-IR *	1.7 (1.2–3.2)	2.2 (1.4–4.5)	2.4 (1.6–3.4)	0.272
Gestational age (weeks)	39 (39–40)	39 (38–40)	39 (38–40)	0.599
Vaginal delivery *n* (%)	27 (69.2)	40 (80)	84 (72.3)	0.480
Birth weight (g)	2598 ± 279 ^a^	3268 ± 270 ^c^	3929 ± 251 ^b^	<0.001
Male sex *n* (%)	22 (56.4)	21 (42)	27 (57.4)	0.241
Fat mass (%)	6.8 ± 3.6 ^a^	11.6 ± 2.4 ^c^	12.0 ± 2.0	<0.001
Ponderal Index (g/cm^3^)	2.5 ± 0.2 ^a^	2.8 ± 0.3 ^c^	2.9 ± 0.2 ^b^	<0.001
Cb insulin (mcUI(mL)	2.5 (1.1–4.3)	4.3 (2.2–4.7)	8.2 (6.2–14.1) ^b^	<0.001
**Cord Blood ^1^H-NMR-Assessed Lipoprotein Profile**				
	**SGA (*n* = 39)**	**AGA (*n* = 50)**	**LGA (*n* = 49)**	***p*-value**
VLDL-cholesterol (mg/dL)	8.5 ± 3.6 ^b^	8.1 ± 3.5 ^c^	5.8 ± 3.3	0.001
IDL-cholesterol (mg/dL)	5.1 ± 1.4	4.8 ± 2.0	4.5 ± 1.7	0.353
LDL-cholesterol (mg/dL)	66.3 ± 11.3 ^a,b^	72.1 ± 9.5	72.0 ± 7.1	0.005
HDL-cholesterol (mg/dL)	37.3 ± 6.6 ^a^	44.2 ± 8.7 ^c^	40.1 ± 8.5	<0.001
VLDL-triglycerides (mg/dL)	34.1 ± 10.1 ^a,b^	29.0 ± 8.2	27.5 ± 8.3	0.002
IDL-triglycerides (mg/dL)	4.7 ± 1.3	4.8 ± 1.7	4.1 ± 1.6	0.092
LDL-triglycerides (mg/dL)	4.5 ± 1.7	4.6 ± 2.1	4.3 ± 1.7	0.797
HDL-triglycerides (mg/dL)	7.4 ± 3.6 ^a^	10.1 ± 3.2 ^c^	7.0 ± 3.8	<0.001
VLDL-P (nmol/L)	26.1 ± 7.1 ^b^	23.6 ± 6.1	21.2 ± 6.6	0.003
Large VLDL-P (nmol/L)	0.8 ± 0.3 ^b^	0.7 ± 0.3	0.6 ± 0.3	0.001
Medium VLDL-P (nmol/L)	2.0 (0.9–2.7) ^a,b^	0.7 (0.5–1.5)	1.1 (0.6–1.5)	0.001
Small VLDL-P (nmol/L)	23.20 ± 5.71 ^b^	21.68 ± 4.82	19.35 ± 5.84	0.005
LDL-P (nmol/L)	455.7 ± 79.7 ^b^	504.0 ± 62.6	493.8 ± 50.8	0.002
Large LDL-P (nmol/L)	86.2 ± 12.6 ^b^	94.1 ± 10.6	97.0 ± 10.9	<0.001
Medium LDL-P (nmol/L)	67.5 ± 32.1	59.0 ± 44.6	60.6 ± 35.7	0.555
Small LDL-P (nmol/L)	302.1 ± 48.5	350.9 ± 37.4	336.9 ± 41.3	<0.001
HDL-P (nmol/L)	15.8 ± 3.7 ^a,b^	19.9 ± 4.1 ^c^	16.7 ± 3.5	<0.001
Large HDL-P (nmol/L)	0.4 ± 0.1 ^a^	0.3 ± 0.1	0.3 ± 0.1	0.374
Medium HDL-P (nmol/L)	9.5 ± 1.2	10.0 ± 1.4	9.9 ± 1.6	0.194
Small HDL-P (nmol/L)	6.0 ± 3.9 ^a^	9.6 ± 3.6 ^c^	6.5 ± 3.0	<0.001

Data are presented as mean ± SD and median (IQR, 25–75) for parametric and nonparametric variables, respectively. Statistical analysis included ANOVA followed by Bonferroni for post hoc test. M: maternal blood; Cb: cord blood; BMI: body mass index; HOMA-IR: Homeostatic Model Assessment for Insulin Resistance; LDL: low-density lipoproteins; HDL: high-density lipoproteins; VLDL: very low-density lipoproteins; IDL: intermediate-density lipoproteins; VLDL-P: VLDL number of particles; IDL-P: IDL number of particles; LDL-P: LDL number of particles; HDL-P: HDL number of particles; SGA: small for gestational age; AGA: appropriate for gestational age; LGA: large for gestational age. * Parameters measured in maternal blood obtained in gestational weeks 33–36. Only 70 control and 51 gestational diabetes mellitus women were included in the analysis. ^a^
*p* < 0.05 SGA vs. AGA; ^b^
*p* < 0.05 SGA vs. LGA; ^c^
*p* < 0.05 AGA vs. LGA.

**Table 3 biomedicines-10-01033-t003:** Cord blood lipid profile determined by 1H-NMR-based methods across the birth-weight categories in GDM and control women.

	Control			GDM			*p*-Values		
	SGA (*n* = 25)	AGA (*n* = 25)	LGA (*n* = 24)	SGA (*n* = 14)	AGA (*n* = 25)	LGA (*n* = 23)	GDM	BW	Interaction
VLDL-cholesterol (mg/dL)	9.0 ± 3.8	7.0 ± 2.6	5.5 ± 2.4	7.6 ± 3.1	9.1 ± 3.9	6.1 ± 3.9	0.474	0.001	0.06
IDL-cholesterol (mg/dL)	5.3 ± 1.5	4.1 ± 1.5 a	4.6 ± 1.5	4.8 ± 1.6	5.7 ± 2.2 a	4.4 ± 2.0	0.338	0.403	0.006
LDL-cholesterol (mg/dL)	66.7 ± 12.8	70.2 ± 8.1	73.6 ± 7.1	65.6 ± 8.8	74.1 ± 9.7	70.2 ± 5.8	0.907	0.005	0.135
HDL-cholesterol (mg/dL)	37.5 ± 7.2	44.5 ± 8.8	40.7 ± 9.2	36.9 ± 5.5	43.8 ± 8.8	39.5 ± 7.9	0.579	0.001	0.983
VLDL-triglycerides (mg/dL)	34.0 ± 11.1	26.4 ± 5.8	27.8 ± 5.9	34.3 ± 8.5	31.7 ± 9.6	27.3 ± 10.3	0.263	0.002	0.223
IDL-triglycerides (mg/dL)	4.9 ± 1.4	4.1 ± 1.2 a	4.2 ± 1.2	4.3 ± 1.3	5.5 ± 1.8 a	4.1 ± 1.9	0.440	0.080	0.005
LDL-triglycerides (mg/dL)	4.8 ± 1.8	3.9 ± 1.7 a	4.6 ± 1.9	3.9 ± 1.3	5.1 ± 2.2 a	3.9 ± 1.3	0.735	0.749	0.007
HDL-triglycerides (mg/dL)	7.5 ± 3.4	9.6 ± 3.6	6.5 ± 3.4	7.2 ± 3.9	10.7 ± 2.6	7.2 ± 3.9	0.351	<0.001	0.651
VLDL-P (nmol/L)	26.3 ± 7.8	21.8 ± 4.7	21.1 ± 4.5	25.8 ± 5.9	25.6 ± 6.8	21.3 ± 8.4	0.307	0.004	0.244
Large VLDL-P (nmol/L)	0.8 ± 0.3	0.6 ± 0.3	0.6 ± 0.2	0.8 ± 0.3	0.8 ± 0.3	0.6 ± 0.3	0.217	0.001	0.361
Medium VLDL-P (nmol/L)	2.2 ± 1.5	0.9 ± 0.6 a	1.3 ± 0.9	1.9 ± 1.2	1.7 ± 1.5 a	1.2 ± 0.8	0.452	0.002	0.038
Small VLDL-P (nmol/L)	23.3 ± 6.3	20.3 ± 5.1	19.2 ± 3.8	23.1 ± 4.9	23.1 ± 5.2	19.5 ± 7.5	0.312	0.006	0.365
LDL-P (nmol/L)	461.4 ± 90.5	488.1 ± 57.2	505.9 ± 57.3	446.1 ± 59.3	520.6 ± 64.9	481.2 ± 40.3	0.823	0.001	0.070
Large LDL-P (nmol/L)	84.4 ± 12.9	94.2 ± 10.0	99.1 ± 11.6	89.0 ± 11.9	94.0 ± 11.3	94.8 ± 9.7	0.998	<0.001	0.200
Medium LDL-P (nmol/L)	73.8 ± 34.2	48.9 ± 39.8 a	65.2 ± 40.4	56.9 ± 25.8	71.6 ± 46.7 a	55.7 ± 30.1	0.931	0.0740	0.021
Small LDL-P (nmol/L)	303.2 ± 54.4	347.0 ± 38.8	341.6 ± 45.6	300.2 ± 38.2	355.0 ± 36.4	330.7 ± 36.4	0.794	<0.001	0.553
HDL-P (nmol/L)	16.6 ± 3.8	20.0 ± 4.2	16.8 ± 3.7	14.6 ± 3.4	19.8 ± 4.1	16.7 ± 3.4	0.269	<0.001	0.442
Large HDL-P (nmol/L)	0.3 ± 0.1	0.3 ± 0.1	0.3 ± 0.1	0.4 ± 0.1	0.3 ± 0.1	0.3 ± 0.1	0.048	0.159	0.145
Medium HDL-P (nmol/L)	9.1 ± 1.1	9.9 ± 1.3	10.0 ± 1.8	10.0 ± 1.0	10.0 ± 1.5	9.8 ± 1.5	0.251	0.394	0.193
Small HDL-P (nmol/L)	7.1 ± 3.9	9.8 ± 3.6	6.4 ± 3.3	4.2 ± 3.3	9.5 ± 3.7	6.6 ± 2.8	0.089	<0.001	0.101

Data are presented as mean ± SD and median (IQR, 25–75) for parametric and nonparametric variables, respectively. Statistical analysis included two-way ANOVA followed by Bonferroni post hoc test. A letter “a” indicates the lipoprotein profile determinations responsible for the observed interaction. GDM: gestational diabetes mellitus; Cb: cord blood; VLDL: very low-density lipoproteins; IDL: intermediate-density lipoproteins; LDL: low-density lipoproteins; HDL: high-density lipoproteins; VLDL-P: VLDL number of particles; IDL-P: IDL number of particles; LDL-P: LDL number of particles; HDL-P: HDL number of particles.

**Table 4 biomedicines-10-01033-t004:** Adjusted odds ratio for the association between large and small LDL particles with the development of obesity at 2 years of age.

	Model R2	Exp (B)	95% CI for Exp (B)	*p*-Value
Large LDL-P * (nmol/L)	0.234	1.052	0.998–1.109	0.058
Small LDL-P * (nmol/L)	0.251	1.018	1.002–1.034	0.023

Logistic regression analysis. * Adjusted for gestational diabetes mellitus, gestational age at delivery, birth weight, sex, pre-gestational BMI and gestational weight gain. LDL-P: LDL particles; CI: confidence interval.

## Data Availability

Data are available from the authors upon reasonable request.

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
