# Peer review of "Cord Blood Advanced Lipoprotein Testing Reveals an Interaction between Gestational Diabetes and Birth-Weight and Suggests a New Early Biomarker of Infant Obesity"

_biomedicines, 2022, doi:10.3390/biomedicines10051033_

Round 1

Reviewer 1 Report

In the present study, Algaba-Chueca and colleagues investigated the umbilical cord serum lipoprotein profile in small for gestational age (SGA), appropriate for gestational age (AGA) and large for gestational age (LGA) neonates from Gestational Diabetes Mellitus (GDM) and control pregnancies. Moreover, they performed a 2-year follow-up study on 103 children. They demonstrated that  GDM modified umbilical cord blood lipoprotein profile in AGA neonates and that an altered fetal lipoprotein pattern is associated with obesity development at 2 years. The Authors concluded that fetal lipoprotein profile might be an early biomarker for the development of later diseases
This is a very interesting study addressing a novel issue as fetal lipoprotein profile as biomarkers for obesity development. Thus, it is likely to be of great interest to the readers of Biomedicines.
However, there are several points that the Authors must address before publication
  1. The Authors should explain the following abbreviations: NMR, SGA, AGA and LGA, IDL, VLDL, LDL. Moreover, the Authors stated that they included 74 controls and 62 GDM pregnant women (n=136) and 39 SGA, 50 AGA and 49 LGA fetuses (n=138). Did you included also twin pregnancies? Please, clarify.
  2. Please, add the definition of AGA and controls. It is not clear if in the control populations you included obese and/or GDM pregnancies. Please, clarify.
  3. Another concern is about the GDM population. You included GDM patients in diet and insulin therapy in the same group (n=31). However, the Authors should analyze them as two different groups. Please, clarify.
  4. The Authors stated that “Infant data included sex, gestational age, way of delivery….”. However, no information were reported in the study population and in the Results section. Did you find differences between male and female neonates?
  5. GDM development is often associated with obesity. How many obese women did you included in your study population? Can you please add gestational BMI other that pre-gestational BMI in Table 1 and 2?
  6. Table 2 and 3, Figure 1. When the Authors found a significant difference among more than two groups, they should perform comparisons of one group with each other in order to identify which is significantly different by the other and add symbols in tables and figures.
  7. The Authors performed a follow-up study on 103 children (78 normal weight and 25 obese). It is not clear if the 78 normal weight children were associated to control mothers and the 25 obese children were associated to GDM mothers. Please clarify.
  8. The first sentence of the discussion section is a repetition of the results. Please remove.
  9. In the discussion section the Authors stated: “GDM offspring belonging to the LGA and SGA groups have a lipoprotein profile more similar to controls”. Did you mean more similar to LGA and SGA babies from control pregnant women? Please, clarify.
  10. Please rephrase the sentence “When analyzing the lipoprotein profile according to birth-weight categories (Table 2), most of the differences observed in the whole group were replicated both, in GDM and control women.” in order to make it more clear.
  11. Discussion section. Please clarify why your findings may support an effect of fetal growth accretion instead of the glucose status.
  12. The Authors concluded that postnatal insulin resistance may be programmed in utero. Did you consider the role of the placenta? You should mentioned it in the discussion section.

Author Response

Response to Reviewer 1

In the present study, Algaba-Chueca and colleagues investigated the umbilical cord serum lipoprotein profile in small for gestational age (SGA), appropriate for gestational age (AGA) and large for gestational age (LGA) neonates from Gestational Diabetes Mellitus (GDM) and control pregnancies. Moreover, they performed a 2-year follow-up study on 103 children. They demonstrated that GDM modified umbilical cord blood lipoprotein profile in AGA neonates and that an altered fetal lipoprotein pattern is associated with obesity development at 2 years. The Authors concluded that fetal lipoprotein profile might be an early biomarker for the development of later diseases
This is a very interesting study addressing a novel issue as fetal lipoprotein profile as biomarkers for obesity development. Thus, it is likely to be of great interest to the readers of Biomedicines.

We do appreciate the suggestions made by the reviewer since they have prompted us to considerably improve the manuscript.

However, there are several points that the Authors must address before publication

  1. The Authors should explain the following abbreviations: NMR, SGA, AGA and LGA, IDL, VLDL, LDL. Moreover, the Authors stated that they included 74 controls and 62 GDM pregnant women (n=136) and 39 SGA, 50 AGA and 49 LGA fetuses (n=138). Did you included also twin pregnancies? Please, clarify.

As suggested by the reviewer we have included the explanation of the abbreviations for NMR, IDL, LDL and VLDL in the abstract. After reviewing the main manuscript, we have verified that they were already explained in the introduction or in the methodology section. The text has been highlighted in yellow.

All women included had a singleton pregnancy as specified in point 1 of the inclusion criteria in the methodology section. It has also been highlighted in yellow.

  1. Please, add the definition of AGA and controls. It is not clear if in the control populations you included obese and/or GDM pregnancies. Please, clarify.

The AGA group included infants born with a birth-weight between the tenth and ninetieth percentiles. We have changed the definition of AGA to make it easier to understand. It is underlined in read in the Clinical and Demographic Data subsection.

Pregnant women were assigned to the control group if they had normal glucose tolerance during gestation and to the GDM group if they had two abnormal results in the 100-gr glucose tolerance test. This information was already present in the second paragraph of the Study Subjects subsection. It has been highlighted in yellow.

Both groups included normal weight, overweight and obese women. The distribution of maternal weight groups was similar in women with GDM and controls (p=0.307). A sentence with this information has been included in the Results section.

In the control group, 42 pregnant women were normal weight, 18 overweight and 14 obese. In the GDM group, 27 pregnant women were normal weight, 20 overweight and 15 obese.

  1. Another concern is about the GDM population. You included GDM patients in diet and insulin therapy in the same group (n=31). However, the Authors should analyze them as two different groups. Please, clarify.

Since treatment with insulin at the end of pregnancy may activate placental nutrient transport to the fetus and promote placental fatty acid transfer, this point was of special concern to us. In case it might be a confounding factor, we assessed differences in the advanced lipoprotein profile and the distribution of infants between the two groups. No differences in the 1HNMR assessed lipoprotein profile between GDM women treated with diet or insulin were observed, except for a lower concentration of medium HDL-P in the insulin treated group (10.3± 1.2 vs 9.6±1.4; p=0.026), as it was stated in the results section. Also, a similar distribution according to birth weight categories was found between the diet and insulin-treated group (p=0.321). For this reason, they were all analyzed together.

It is important to keep in mind that only 10% of children born in a normal population fall into the SGA group, and if we take into account that the GDM in our population is around 6-8%, it is difficult to get a sample of more than 15 small-for-gestational-age children in each of the groups. For this reason, we checked that there were no differences between the groups in terms of weight distribution and the main results of the lipoprotein pattern.

A comment referring to this limitation has been included in the Discussion section.

  1. The Authors stated that “Infant data included sex, gestational age, way of delivery….”. However, no information were reported in the study population and in the Results section. Did you find differences between male and female neonates?

The reviewer is right. The number and percentage of male infants and vaginal deliveries have been added to tables 1 and 2. No statistical differences were observed in infant sex or the way of delivery between the groups. Gestational age was already included in tables 1 and 2.

  1. GDM development is often associated with obesity. How many obese women did you included in your study population? Can you please add gestational BMI other that pre-gestational BMI in Table 1 and 2?

The distribution of women with normal weight, overweight and obesity has been added to the Results section, as addressed in #1.

As suggested by the reviewer, we have also added the final BMI in tables 1 and 2. A description of this new variable has been added to the Clinical and Demographic Data subsection. Parallel to weight gain, differences were observed between birth-weight groups, with an increase in mean BMI relative to birth-weight. In the Bonferroni post-hoc analysis, significant differences were observed between the SGA and LGA groups, indicated by the letter “b” in table 2. A comment has also been included in the Results section.

  1. Tables 2 and 3, Figure 1. When the Authors found a significant difference among more than two groups, they should perform comparisons of one group with each other in order to identify which is significantly different by the other and add symbols in tables and figures.

This information was already included in table 2. The letters “a”, “b” and “c” next to the means ± SD in the columns corresponding to SGA and AGA showed the differences observed between the different groups after the Bonferroni post-hoc study. The corresponding explanation was shown in the legend.

In table 3 we have included all the lipoprotein profile determinations separated according to the presence or not of GDM and birth-weight categories to show the interaction between these two variables. We have added a letter “a” to indicate the lipoprotein profile determinations responsible for the observed interaction, after Bonferroni post-hoc test. Also in the figure 1 we have included a line indicating which variables showed the interaction.

The differences observed between the groups according to birth-weight are shown in table 2.

  1. The Authors performed a follow-up study on 103 children (78 normal weight and 25 obese). It is not clear if the 78 normal weight children were associated to control mothers and the 25 obese children were associated to GDM mothers. Please clarify.

Of the 103 children followed up, 78 were normal weight and 25 were obese at 2 years. Of the 78 children with normal weight, 44 were children born to control mothers and 34 were children born to mothers with GDM, while in the group of obese children, only 8 were children born to control mothers and 17 were children born to mother with diabetes (p=0.034). For this reason, obesity was more frequent in children born to mothers with GDM, mothers with obesity or mothers with higher birth-weight as was already stated in subsection 3.4 entitled: “Cord blood 1H-NMR-lipoprotein Profile is associated with Obesity at two years”.

  1. The first sentence of the discussion section is a repetition of the results. Please remove.

Done.

  1. In the discussion section the Authors stated: “GDM offspring belonging to the LGA and SGA groups have a lipoprotein profile more similar to controls”. Did you mean more similar to LGA and SGA babies from control pregnant women? Please, clarify.

We appreciate the reviewer’s comments. It was unclear. We have rephrased the sentences.

  1. Please rephrase the sentence “When analyzing the lipoprotein profile according to birth-weight categories (Table 2), most of the differences observed in the whole group were replicated both, in GDM and control women.” in order to make it more clear.

This sentence has been changed.

  1. Discussion section. Please clarify why your findings may support an effect of fetal growth accretion instead of the glucose status.

As in the previous comments, we agree that this sentence was not clear and we have rewritten it.

  1. The Authors concluded that postnatal insulin resistance may be programmed in utero. Did you consider the role of the placenta? You should mentioned it in the discussion section.

Yes, we are aware of the key role that the placenta plays in nutrient handling and fetal development. Since in this study we have not explored the role of the placenta in the passage of fatty acids or in lipid metabolism, we have not mentioned it.

However, since it is a determining factor in the passage of nutrients, which may be responsible for growth disturbances, we have introduced a comment on this in the discussion

Reviewer 2 Report

In this manuscript authors used a novel NMR-based approach to profile the umbilical cord serum lipoproteins and found that size, lipid content, number and concentration of particles within their subclasses were similar between offspring born to control and GDM mothers. Moreover, authors found an interaction between GDM and birth-weight categories for IDL-cholesterol content and IDL- and LDL-triglyceride content, and the number of medium VLDL and LDL particles specifically in AGA neonates. In addition, in a 2-year follow-up study they found that small LDL particles were independently associated with offspring obesity at two years demonstrating that GDM alters triglyceride and cholesterol lipoprotein content in AGA neonates born from GDM mothers displaying a profile more similar to adults with dyslipidemia. 

The manuscript is clear and generally well written. The cohort studied has a good size and inclusion/escusion criteria are appropriated for this typer of study. To my opinion this manunuscript can be accepted in the present form. 

Author Response

Thank you very much for your comment.

Round 2

Reviewer 1 Report

The Authors addressed all Reviewer's requests.